# Microstructure and Texture Evolution in Low Carbon and Low Alloy Steel during Warm Deformation

**DOI:** 10.3390/ma15072702

**Published:** 2022-04-06

**Authors:** Sheng Xu, Haijie Xu, Xuedao Shu, Shuxin Li, Zhongliang Shen

**Affiliations:** 1College of Mechanical Engineering and Mechanics, Ningbo University, Ningbo 315211, China; xs@zbti.edu.cn (S.X.); xuhaijie@nbu.edu.cn (H.X.); lishuxin@nbu.edu.cn (S.L.); 2Department of Mechanical Engineering, Zhejiang Business and Technology Institute, Ningbo 315012, China; szl@zbti.edu.cn

**Keywords:** low carbon and low alloy steel, warm deformation, texture, recrystallization

## Abstract

Warm compression tests were carried out on low carbon and low alloy steel at temperatures of 600–850 °C and stain rates of 0.01–10 s^−1^. The evolution of microstructure and texture was studied using a scanning electron microscope and electron backscattered diffraction. The results indicated that cementite spheroidization occurred and greatly reduced at 750 °C due to a phase transformation. Dynamic recrystallization led to a transition from {112}<110> texture to {111}<112> texture. Below 800 °C, the intensity and variation of texture with deformation temperature is more significant than that above 800 °C. The contents of the {111}<110> texture and {111}<112> texture were equivalent above 800 °C, resulting in the better uniformity of γ-fiber texture. Nucleation of <110>//ND-oriented grains increased, leading to the strengthening of <110>//ND texture. Microstructure analysis revealed that the uniform and refined grains can be obtained after deformation at 800 °C and 850 °C. The texture variation reflected the fact that 800 °C was the critical value for temperature sensitivity of warm deformation. At a large strain rate, the lowest dislocation density appeared after deformation at 800 °C. Therefore, 800 °C is a suitable temperature for the warm forming application, where the investigated material is easy to deform and evolves into a uniform and refined microstructure.

## 1. Introduction

In industrial conditions, more than 80% of energy is consumed in heating and the rest in rolling [1]. Taking the advantages of cold forming and hot forming, warm forming at a temperature of 650–850 °C was applied in industry as an energy-saving technology. The behavior and microstructure evolution of warm deformation have been studied on various carbon steels. Typically, recovery and recrystallization occurred simultaneously and interacted with each other during deformation, and their speed and share in microstructural variation depended on the chemical composition, initial microstructure and process parameters [2]. At low deformation temperature and high strain rate, the work softening rate increased significantly [3], and the increase in carbon content led to a decrease in deformation activation energy [4].

Grain refinement occurred in both medium carbon and low carbon alloy steels during warm deformation [5,6,7]. Ultra-fine grain microstructure also can be obtained by warm deformation of ultra-low carbon steel [8,9]. Continuous dynamic recrystallization (CDRX) can lead to the formation of new fine ferrite grains [7,10]. The cementite precipitated at the ferrite boundaries, due to intragranular nucleation activation, then caused ferrite to nucleate over the α/γ interface [11]. The elongated ferrite grains continuously dynamically recrystallize to form the equiaxed fine ferrite grains [12]. Eghbali [13,14] conducted extensive research on the warm deformation of low carbon steels and discovered that strain rate had an important effect on grain refinement in the CDRX process.

Grain refinement was also directly related to the spheroidization of cementite during warm deformation [15,16]. Warm deformation at 650–750 °C promoted the spheroidization of carbides, leading to flow softening [17,18]. In addition, the spheroidization of pearlite was accelerated due to the heavy warm deformation of ferritic-pearlitic carbon steel, resulting in the formation of completely spheroidized cementite, and the deformation at 670–700 °C led to homogeneous distribution of cementite particles [19].

It is well recognized that texture is one of the key factors affecting the properties of BCC metals and alloys [20], and texture might have negative effects on material characteristics [20,21], such as ridging, earring, etc. A large number of scholars paid attention to the grain orientation for different annealing processing parameters of steel sheets [22], so as to obtain large plastic strain ratios. In general, {111}// ND has a larger Taylor factor than {100}//ND in BCC metals, which means that many slip systems are activated and grain fragmentation is easily promoted in {111}//ND [23]. The texture of {001}<110> and {111}<110> developed significantly, and the fraction of the high angle boundary increased with the equivalent strain in an ultra-low carbon steel compression [24]. The {111} orientations were the first to recrystallize while the α-fiber was present until the end of recrystallization [25].

At present, there have been a large number of studies on warm deformation of carbon steel. In most previous studies, steel sheets were investigated to obtain blank material with better plasticity for further cold processing [26], such as drawing or stamping. In fact, the warm rolling or warm cross wedge rolling (WCWR) of shaft parts and warm extrusion of auto parts are also widely applied in industry [27,28,29]. Bulzak [1] suggested that WCWR consumed less energy to heat the workpiece than hot rolling. In contrast, Huang [29] proposed that the warm rolling torque was three times that of the hot rolling torque and, although the heating energy consumption was reduced, the spheroidization of cementite and the recrystallization of ferrite had remarkable effects on the tensile strength and yield strength. In addition, most of the experimental materials in previous studies were heated to complete austenitization then cooled to the ferrite-austenite region and ferrite region for compression testing [30]. In fact, the material is directly heated to the forming temperature in the industrial applications. Even if the final temperature is the same, the microstructure evolution from ferrite to austenite (heating process) is significantly different to the evolution from austenite to ferrite (cooling process). Thus, an appropriate forming temperature is provided by warm rolling, especially in heavy deformation, and determines the energy consumption, formability and microstructure that influences the mechanical properties of products. Moreover, investigating the temperature sensitivity of the material, the influence of process parameters on microstructure evolution, and the homogeneity of plastic deformation, is helpful and essential to optimize the technical parameters (temperature, strain rate, area reduction, etc.) to accomplish the more effective forming qualities with fewer resources [31].

In this study of warm deformation, low carbon and low alloy steel (20CrMoA) with high quenching ability, good machinability, cold strain plasticity, and which is widely used in shafts and gears [32], was investigated as an experimental material. The evolution of microstructure and texture during warm deformation under different parameters was studied. Meanwhile, it is considered that the warm forming process in industrial applications mostly involves heavy deformation at high strain rate. The microstructure at high strain rate is thoroughly analyzed to determine the proper deformation temperature and provide a reference for process application.

## 2. Material and Experimental Procedure

The chemical composition of the studied low carbon and low alloy structural steel was 0.2%C, 0.24%Si, 0.52%Mn, 0.92%Cr, 0.16%Mo (in wt%). The initial microstructure of the low carbon steel was lamellar with alternate formations of ferrite and pearlite. The cylindrical specimens, with a diameter of 8 mm and a height of 12 mm sampled from a homogenized bar, were compressed on a Gleeble-3500 thermal simulator. In this experiment, the deformation temperature was 600, 650, 700, 750, 800 and 850 °C. Each specimen was heated to the specified deformation temperature at a rate of 10 °C/s and held for 3 min under isothermal conditions for heat balance. The specimen was compressed along the axial direction (ND) at strain rates of 0.01, 0.1, 1, and 10 s^−1^ with the deformation ratio of 70% as shown in Figure 1. The specimens were immediately quenched in water to maintain the microstructure after compression. After that, the compressed specimens were sectioned, polished, and etched to obtain the microstructures under different compression conditions. Micro-textures were examined by electron backscattered diffraction (EBSD), and microstructures were observed using a scanning electron microscope (SEM). The percentages of high-angle grain boundaries (HAGBs) with the misorientation angles higher than 15° and low-angle grain boundaries (LAGBs) with the misorientation angles between 2° and 15° were calculated [33,34].

The EBSD map of initial microstructure was shown in Figure 2. The uniform microstructure consisting of equiaxed grains can be observed with the average size of the grains of 10.74 μm. The transformation temperature was calculated by JmatPro software (Version 7.0, Sente Software Ltd., Guildford, Surrey, UK) based on the chemical composition of the experimental steel.

## 3. Results and Discussion

### 3.1. Microstructures Evolution

According to the results calculated by JmatPro software, the start equilibrium transformation temperature AC1 and finish equilibrium transformation temperature Ac3 from ferrite to austenite are determined to be 731.5 °C and 824.7 °C, respectively. Figure 3 shows the low magnification SEM images of different deformation conditions. At 700 °C and 0.01 s^−1^, the microstructure of steel comprises pearlite and ferrite, even though the deformation time is longer (Figure 3a). At 750 °C, apparent transformation occurred and the steel showed the α + γ double-phase region. In this region, recrystallized ferrite and the amount of martensite formed after cooling of austenite were observed as shown in Figure 3b. In addition, a few FeC_3_ particles appeared. With the increasing deformation temperature, the fraction volume of martensite increases, while FeC_3_ particles were not observed in the microstructure (Figure 3c). Moreover, the ferrite grains are smaller. In the case that the deformation temperature exceeds the AC3 temperature, the steel is completely austenitized (Figure 3d).

To distinguish details in the microstructures, higher-magnification SEM images in different deformation conditions are provided in Figure 4. Significant fragments and particles are observed as a result of the rupture and spheroidization of cementite. At 650 °C and 0.1 s^−1^ as shown in Figure 4a, cementites are dominated by broken fragments, which is related to the deformation temperature. In the process of warm deformation, the ferrite and cementite in pearlite can deform co-ordinately at small strain. During heavy deformation, the co-ordinative deformation state will be broken due to significant differences in the mechanical properties between ferrite and cementite. The cementite with a poor plastic property is prone to bending, melting and spheroidization. When heavy deformation compression is carried out at low temperature, the cementite is fractured under a strong plastic deformation force, and the fragments are in heterogeneous nucleation. At high deformation temperature, the lamellar cementite gradually dissolves and shrinks to short rods, driven by interface energy. Therefore, more pieces of short rod cementite appeared after the compression at 700 °C, as shown in Figure 4b. The main mechanism is that a large number of dislocations are produced in the ferrite during warm deformation, providing a channel for the rapid diffusion of carbon atoms. Subsequently, the increase of deformation temperature further intensifies the diffusion, causing the concentration gradient of carbon. The lamellar cementites dissolved in the ferrite, under the action of deformation force and thermal effect. When carbon was supersaturated in ferrite, the fine cementite particles precipitated.

As shown in Figure 4c,d, the cementite spheroidized at the boundary between the recrystallized ferrite grain and martenties. In addition, the spheroidization of cementite is greatly reduced in the process of deformation at 750 °C, whether the strain rate is low or high. This is related to the phase transformation in the steel, i.e., the ferrite transferred into austenite at 750 °C. In general, austenite nucleates at the interface of ferrite and cementite. Because a lot of cementites were dissolved by the diffusion of austenite, the spheroidized cementite particles decrease greatly during the deformation. While the undissolved cementite particles remain at the junction of ferrite and austenite which is transformed into martentie after cooling.

### 3.2. Recrystallization Behavior

The EBSD maps of substructure, recrystallized and deformed grains after deformation at 600, 650, and 700 °C (ferrite region) are shown in Figure 5. The blue regions represent recrystallized grains, and the red regions indicate that the grains have undergone plastic deformation and stored distortion energy. The yellow regions refer to the substructures that were not completely recrystallized, and the energy stored in those regions is lower than that in the red regions. In the ferrite region, the EBSD maps of the microstructures were basically the same, and mostly contained deformed grains. Dynamic recrystallization (DRX) of ferrite did not readily occur and dynamic recovery (DRV) was the main softening mechanism, due to the high stacking fault energy (SFE) of the BCC structure, which was susceptible to dislocation climb and cross slip [35]. According to the previous study [36], the width of extended dislocation is small in high SFE materials, which commonly leads to the clustered imperfect dislocations. During heat deformation, dislocation climb and cross slip easily proceeded, resulting in sufficient dynamic recovery. However, the remaining stored energy is insufficient to facilitate the dynamic recrystallization.

The fractions of recrystallized grains, substructure, and deformed grains are shown in Figure 6a, and the fraction of formed grains exceeds 84% under all deformation conditions. Therefore, LAGBs dominate in the misorientation angle, accounting for more than 58% as shown in Figure 5c,d,g,h. The higher proportion of HAGBs can be attributed to the higher volume fraction of DRX [37]. In addition, a low proportion of HAGBs and recrystallized grains indicates that only partial dynamic recrystallization occurs and the main soft mechanism is DRV.

There is no significant difference in grain size under different deformation conditions, with an average grain size of around 3.1 μm. Figure 6b shows the grain size distribution under different deformation conditions. The proportion of size 0–5 μm exceeds 83%, with grains in the size of 0–2 μm being around 40%, indicating that grains are refined due to the local dynamic recrystallization of ferrite.

Figure 7 shows the EBSD maps of recrystallized, substructure, and deformed grains after deformation at 750, 800, and 850 °C. Figure 8a shows the fraction of recrystallized grains, substructure and deformed grains under different deformation conditions, and Figure 8b presents the distribution of grain sizes. As shown in Figure 8a, at a strain rate of 10 s^−1^, the proportion of recrystallized grain after deformation at 850 °C is only 3.73%, which is much less than 32.08% at 800 °C and 5.56% at 650 °C with the strain rate of 0.1 s^−1^. This phenomenon is mainly attributed to phase transformation. Complete austenitizing occurs during deformation at 850 °C due to the fact that the deformation temperature is higher than the AC3 temperature. Consequently, the recrystallized grains at 850 °C are all austenite. However, 850 °C is lower than the complete recrystallization temperature of austenite grains, so only a small amount of dynamic recrystallization occurs in the microstructure.

The volume fraction of recrystallized grain is 32.08% at 800 °C and 10 s^−1^, which is much higher than 14.77% at 800 °C and 1 s^−1^, but the area fraction of substructure is apparently low. There are three main reasons for this phenomenon [38,39]: (i) The deformation time is longer at a low strain rate and the continuous deformation results in the substructures consisting of entangled dislocations in dynamically recrystallized grains. (ii) More severe shear deformation at a large strain rate intensifies dynamic recrystallization to a certain extent. (iii) At large strain rates, most of the deformation heat cannot be dissipated, and is instead stored as heat energy, leading to the temperature rise of the specimen. The higher temperature rise due to a larger strain rate is more favorable for dynamic recrystallization during heavy deformation.

The distributions of misorientation angle and the percentages of HABs reveal the recrystallization behavior shown in Figure 7c,d,g,h. It is worth noting that the proportion of HAGBs is close to 90% at 800 °C and 1 s^−1^, even at a lower percentage of recrystallized grains, indicating that the substructures were not composed of subgrains with LAGBs. DRV excessively consumed the distortion energy at a higher temperature [35,37], so there was a high proportion of both substructures and HAGBs.

Figure 8b illustrates the grain size distribution under different deformation conditions. The proportion of size 0–5 μm also exceeded 82%, In addition, the percentage of grains with the diameter of 0–2 μm decreased, while that in the diameter of 2–5 μm increased, compared with the grains’ size in the ferrite region. Low strain rate provides a longer deformation time, and the recrystallized grains have the opportunity to grow into large sizes. As a result, the average size of grains at 1 s^−1^ is larger than that at 10 s^−1^.

To analyze the dispersion degree of grain distribution, the standard deviation of grain size under each deformation condition was calculated as shown in Figure 9. It can be found clearly that the standard deviation of grain size distribution is smaller when the deformation temperature is higher than 800 °C and the minimum value is 1.08 at 800 °C and 1 s^−1^, indicating that the microstructure with a more uniform grain distribution can be obtained at 800 °C or 850 °C.

### 3.3. Texture Evolution

Figure 10 shows the main texture components and fibers of BCC steels in the φ_2_ = 45° section of the Euler space [40]. Figure 11 shows the orientation maps and micro-texture after deformation at 650 and 700 °C. The grains having blue, red, and green colors are <111>//ND, <001>//ND, <101>//ND orientation, respectively. Under each deformation condition, the microstructure was characterized by elongated grains and mainly consisted of <111>//ND and <001>//ND textures. The intensity of <111>//ND texture is significantly higher than <001>//ND texture. Barnett [41] revealed that {111}<112> grains were nucleated in situ and {111}<110> grains were nucleated at the grain boundaries of deformed {111}<112> grains during recrystallization. In addition, {112}<110> texture transformed to {111}<112>, and {001}<110> transformed to {111}<112> or {111}<110>. Thus, the formation of strong {111}<112> and {111}<110> are attributed to nucleation rate and growth rate [42,43].

As shown in Figure 11, {112}<110> was weak at 650 °C and 0.1 s^−1^ and disappeared at 700 °C and 0.1 s^−1^, indicating that {112}<110> was completely consumed during the recrystallization process, even if the recrystallization proportion was very low. With the increasing temperature or decreasing strain rate, the intensity of <111>//ND texture and {001}<110> (R-Cube) decreased. Meanwhile, the intensity of {001}<110> (Cube) increased under conditions favorable for recrystallization, which indicated that the Cube is recrystallized texture.

Figure 12 shows the orientation maps and micro-texture after deformation at 750, 800 and 850 °C. The microstructures were mainly characterized by elongated grains at 750 °C. After deformation at 800 °C and 850 °C, the equiaxed grains with an average size of 3 μm dynamically recrystallized in the dual-phase deformed microstructures. As in the ferrite region, the texture mainly consisted of <111>//ND and <001>//ND fibers. However, the intensity of each texture component at 750 °C was much stronger than that above 800 °C.

At the fixed strain rate of 1 s^−1^, the intensity of {111}<110> and {001}<100> at 750 °C was higher than that at 700 °C and 800 °C. It indicated that a small amount of austenite has a significant effect on {111}<110> and {001}<100> in the initial stage of phase transformation.

In addition, when the deformation temperature rose from 750 °C to 800 °C, the reduction of texture intensity was much larger than that at the same temperature interval (from 800 °C to 850 °C) and a larger strain rate of 10 s^−1^. It is inferred that there was a critical temperature range between 750 °C and 800 °C, where the texture intensity decreased rapidly with the phase transformation due to rising temperature, and the deformation behavior was more sensitive to the temperature below 800 °C. The steel was in the end stage of phase transformation and austenite was dominant at 800 °C where texture type and intensity variation were similar to austenite, thus the texture intensity was higher than that at 850 °C. It is worth noting that above 800 °C the nucleation of <110>//ND oriented grains increased, leading to the strengthening of <110>//ND texture. 

### 3.4. Effect of Warm Deformation Parameters on Texture Uniformity

Table 1 shows the percentage of the texture component under different deformation conditions. When the deformation temperature reached 800 °C, the content of <111>//ND texture and <001>//ND texture decreased greatly, but there is no significant difference in content between {111}<110> and {111}<112>. Since the stored energy of {111}<112> is higher than that of {111}<110>, {111}<112> preferentially nucleates in the recrystallization process [44], and the content of the components is usually higher than that of {111}<110>. In addition, the transformation rate of {111}<112> oriented grains to {111}<110> orientation is far greater than that of {111}<110> to others [45]. Due to the higher recrystallization proportion above 800 °C, the transformation between {111}<112> and {111}<110> is sufficient, and the intensity of the two components is equivalent. Thus, the uniformity of γ-fiber texture is better above 800 °C. The percentage of <110>//ND texture increased greatly above 800 °C, which shows that the austenite influenced the evolution of <110>//ND texture during warm deformation. At the strain rate of 10 s^−1^ and above 800 °C, the texture of λ-fiber has better uniformity.

### 3.5. Kernel Average Misorientation Distribution

Kernel Average Misorientation (KAM) distribution can reflect the degree of deformation. The blue region possesses the dislocation with the lowest density, while the red region possesses the dislocation with the highest density [44,46]. The grains with low KAM values exhibit a uniform strain distribution, and high values indicate a greater degree of plastic deformation or a higher defect density. From the experience of engineering applications, warm forming such as warm rolling or warm extrusion is heavy deformation under a large strain rate. The KAM maps at large strain rates of 1 s^−1^ and 10 s^−1^ are shown in Figure 13. At strain rate of 1 s^−1^, the dislocation density decreases with the increase of deformation temperature. The sample after deformation at 800 °C has the lowest dislocation density and most homogenous strain distribution. At a fixed strain rate of 10 s^−1^, the dislocation density in the sample deformed at 850 °C looks more uniform, but apparently higher than the others. The sample deformed at 800 °C has a lower dislocation density than the sample with 750 °C, which will show relative stable properties in subsequent processing.

## 4. Conclusions

(1)The spheroidization of cementite is related to deformation temperature. At low temperature, the broken fragments are dominant, while the cementite spheroidizes into small particles and short rods at 700 °C. In the dual-phase region, the cementite is dissolved due to the austenitizing, and the spheroidizing particles decrease rapidly. After deformation at 800 °C, equiaxed grains with the size of 3 μm were observed, which is caused by the dynamic recrystallization of ferrite.(2)In the ferrite region, the texture mainly consists of <111>//ND, <001>//ND fibers. With the increasing temperature or/and the decreasing strain rate, the intensity of {001}<100> (Cube) texture decreases and the {112}<110> texture transforms to {111}<112> texture.(3)800 °C is the critical value for temperature sensitivity of warm deformation. At temperatures below 800 °C, the texture mainly consisted of strong γ-fiber and λ-fiber texture, and the variation of texture strength with deformation temperature is significant. Above 800 °C, the γ-fiber texture shows relatively uniform distribution. The <110>//ND oriented grains appears, leading to the strengthening of <110>//ND texture.(4)Equiaxed grains with smaller average grain size and uniform distribution can be obtained at 800 °C. Under a large strain rate, the fraction of KAM values indicating lower dislocation at 800 °C is much higher than at other deformation temperatures. For this reason, 800 °C is a suitable temperature for warm forming applications.

## Figures and Tables

**Figure 1 materials-15-02702-f001:**
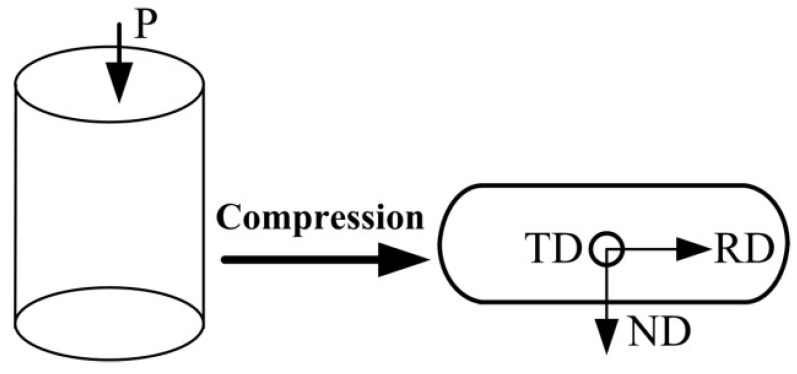
Compression process and the orientations of the specimen.

**Figure 2 materials-15-02702-f002:**
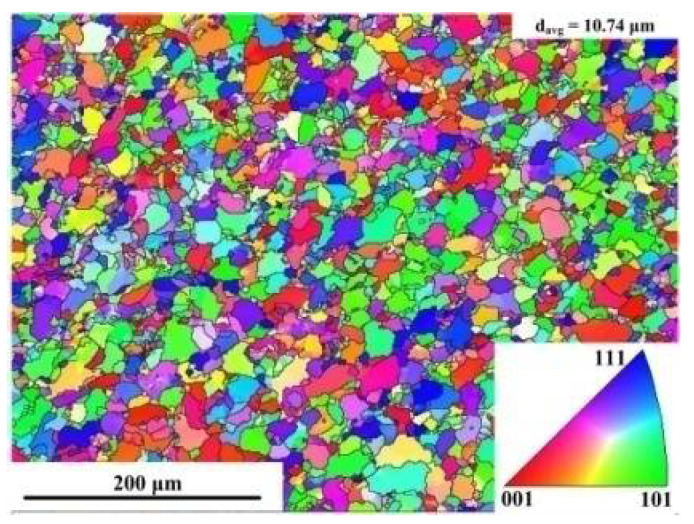
EBSD map of non-deformed microstructure.

**Figure 3 materials-15-02702-f003:**
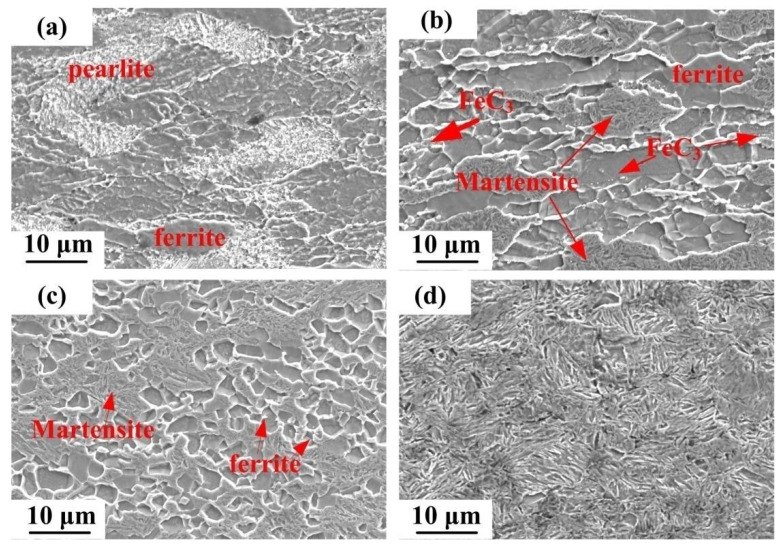
Low magnification SEM images of (**a**) 700 °C, 0.01 s^−1^ (**b**) 750 °C, 0.01 s^−1^ (**c**) 800 °C, 1 s^−1^ (**d**) 850 °C, 1 s^−1^.

**Figure 4 materials-15-02702-f004:**
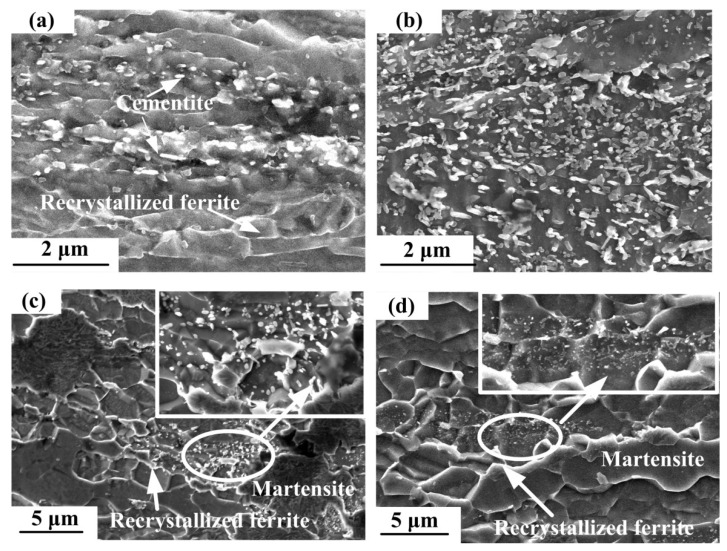
High magnification SEM images of (**a**) 650 °C, 0.1 s^−1^; (**b**) 700 °C, 10 s^−1^; (**c**) 750 °C, 0.1 s^−1^; (**d**) 750 °C, 10 s^−1^.

**Figure 5 materials-15-02702-f005:**
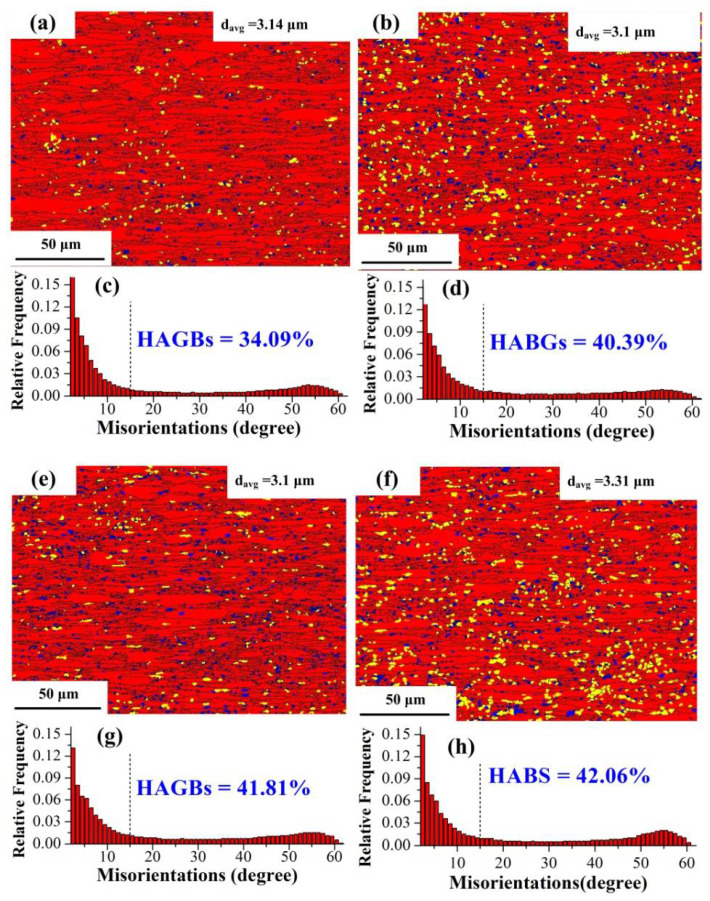
EBSD maps of recrystallized, substructure, and deformed grains; misorientation angle distributions (**a**,**c**) 600 °C, 0.01 s^−1^ (**b**,**d**) 650 °C, 0.1 s^−1^ (**e**,**g**) 700 °C, 0.1 s^−1^ (**f**,**h**) 700 °C, 1 s^−1^.

**Figure 6 materials-15-02702-f006:**
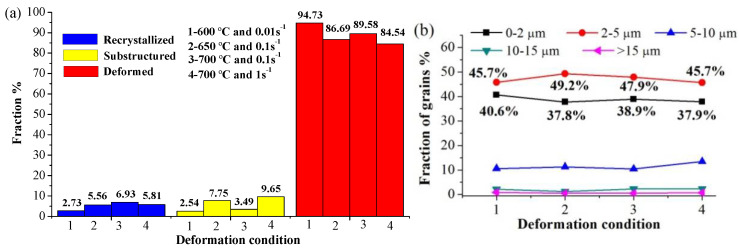
(**a**) The fraction of recrystallized, substructure and deformed grains under different deformation conditions (**b**) Distribution of grain size under different deformation conditions.

**Figure 7 materials-15-02702-f007:**
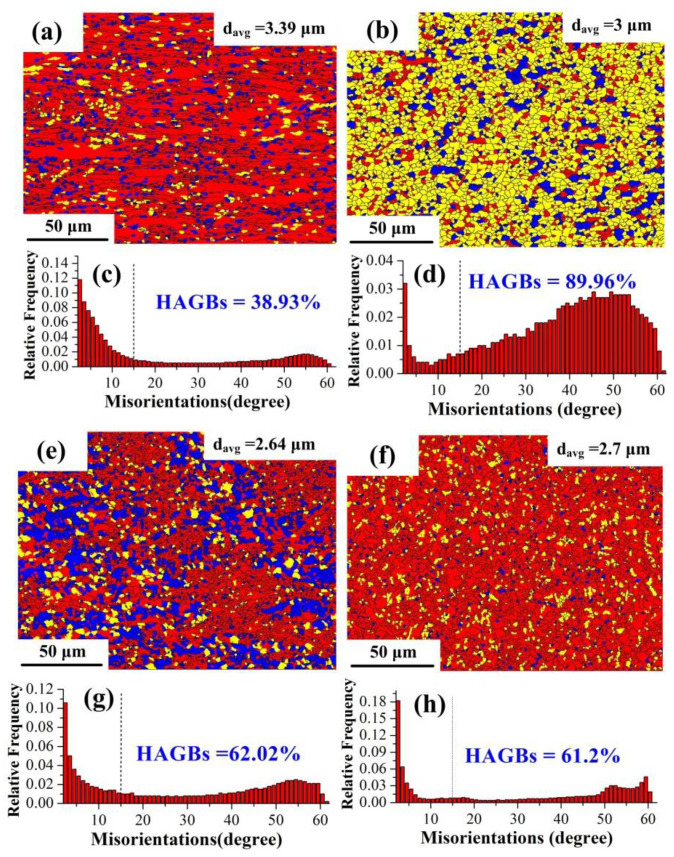
EBSD maps of recrystallized, substructure, and deformed grains, and misorientation angle distributions (**a**,**c**) 750 °C, 1 s^−1^; (**b**,**d**) 800 °C, 1 s^−1^; (**e**,**g**) 800 °C, 10 s^−1^; (**f**,**h**) 850 °C, 10 s^−1^.

**Figure 8 materials-15-02702-f008:**
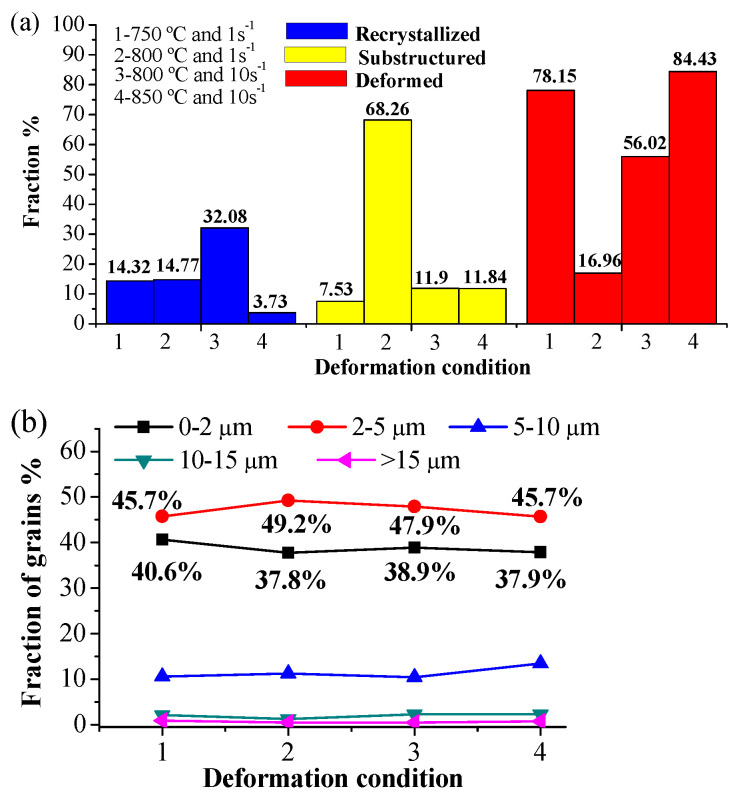
(**a**) The fraction of recrystallized grains, substructure and deformed grains under different deformation conditions, (**b**) Distribution of grain size under different deformation conditions.

**Figure 9 materials-15-02702-f009:**
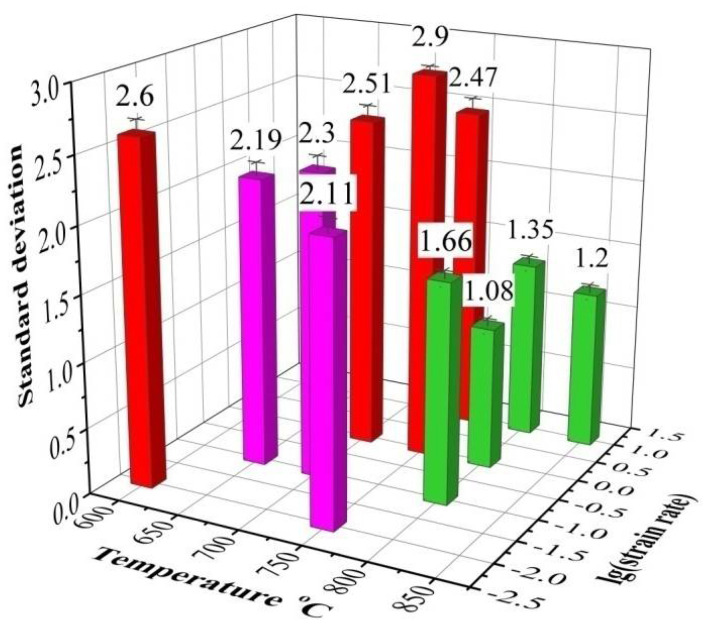
Standard deviation of grain size.

**Figure 10 materials-15-02702-f010:**
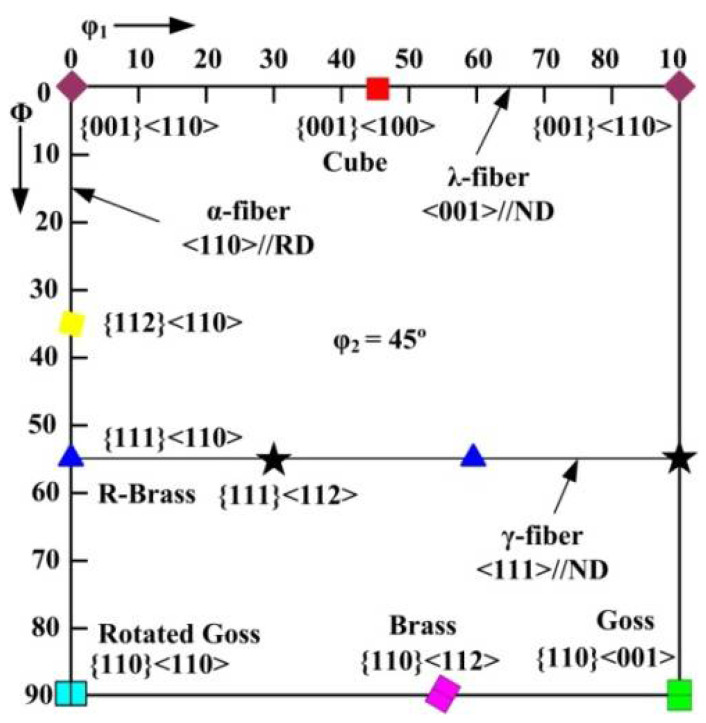
Main texture components and fibers of BCC steels of the Euler space: φ_2_ = 45^°^ section.

**Figure 11 materials-15-02702-f011:**
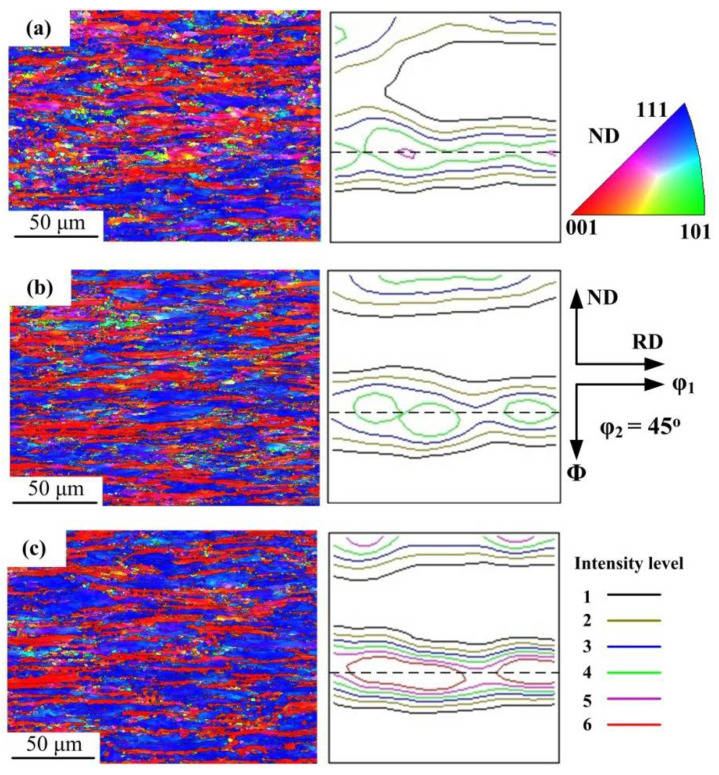
Grain orientation map of (**a**) 650 °C, 0.1 s^−1^ (**b**) 700 °C, 0.1 s^−1^ (**c**) 700 °C, 1 s^−1^.

**Figure 12 materials-15-02702-f012:**
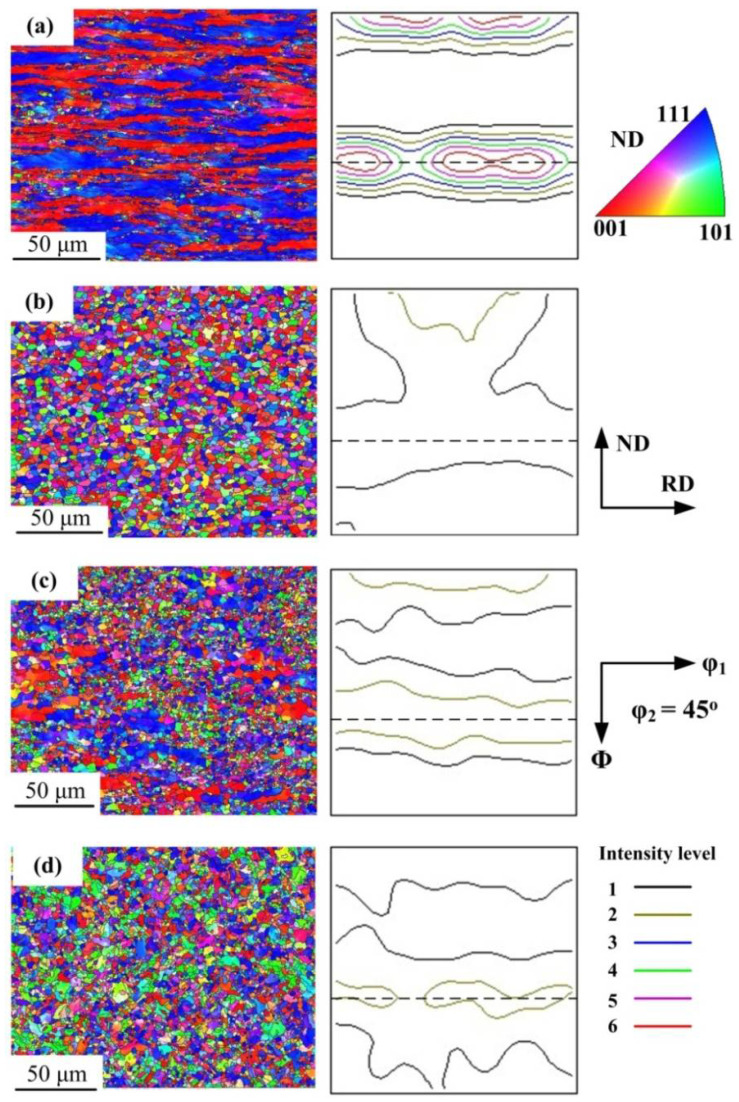
Grain orientation map of (**a**) 750 °C, 1 s^−1^ (**b**) 800 °C, 1 s^−1^ (**c**) 800 °C, 10 s^−1^ (**d**) 850 °C, 10 s^−1^.

**Figure 13 materials-15-02702-f013:**
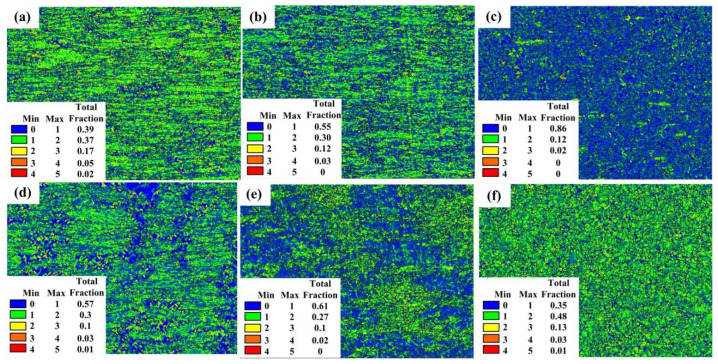
KAM map. (**a**) 700 °C, 1 s^−1^ (**b**) 750 °C, 1 s^−1^ (**c**) 800 °C, 1 s^−1^ (**d**) 750 °C, 10 s>^−1^; (**e**) 800 °C, 10 s^−1^; (**f**) 850 °C, 10 s^−1^.

**Table 1 materials-15-02702-t001:** The percentage of the texture component (divergence angle = 20°).

	{111}	{111}<110>	{111}<112>	{001}	{001}<110>	{001}<100>	{110}
	Texture, %	Component, %	Component, %	Texture, %	Component, %	Component, %	Texture, %
650 °C 0.1 s^−1^	46.1	24.2	26.3	35.1	15.5	9.17	4.96
700 °C 0.1 s^−1^	48.6	24.6	30.2	36.9	10.9	15	5.21
700 °C 1 s^−1^	58.2	32.4	36.7	34.8	15.6	10.2	1.79
750 °C 1 s^−1^	51.1	33.7	27.7	40.4	13.5	17.1	2.64
750 °C 10 s^−1^	49.2	25.5	31.6	40.9	12.6	18.1	3.77
800 °C 1 s^−1^	29.4	14.5	14.3	26.2	4.7	11	21.4
800 °C 10 s^−1^	37.1	18.3	19	26.5	8.48	8.77	18.4
850 °C 10 s^−1^	31.6	15.4	15.6	16.8	6.58	6.68	31.7

## Data Availability

The data presented in this study are available on request from the corresponding author.

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
