# Peer review of "Microstructure and Texture Evolution in Low Carbon and Low Alloy Steel during Warm Deformation"

_materials, 2022, doi:10.3390/ma15072702_

Round 1

Reviewer 1 Report

The manuscript studied microstructure and texture evolution in low carbon and low alloy steel during warm deformation. The manuscript has serious flaws, meaning that this cannot be published in the present form. I recommend major revision, and the following comments should be considered in the revised version. 

1- The abstract is not appropriate, and not representative of the main results. In the abstract, the results should be summarized briefly. 

2- The English language of the manuscript is misleading, and the manuscript is not easy to read for readers. It is recommended to be revised by an English native speaker who is an expert in the field.

3- The introduction should be extended. I could not understand why this study is essential. The motivation should be explained after reviewing the previous related works.

4- In line 88, the definition of LAGB and HAGB were stated. This needs to add references: https://doi.org/10.1016/j.ijhydene.2020.12.028 https://doi.org/10.2320/matertrans.45.3247

5- There are several typos and errors in writing. The manuscript should be proofread before submission!

6- In line 140, the authors explained that dynamic recrystallization (DRX) of ferrite was hard to occur and dynamic recovery (DRV) was the main softening mechanism, due to the high stacking fault energy of BCC structure which was susceptible to dislocation climb and cross slide. What is Cross slide? The right one is cross slip. Please explain why cross slip and climb are easier in high SFE materials.

7- In line 168, three reasons were mentioned for the different extent of recrystallized grains in different strain rates.  (i) The deformation time is longer at a low strain rate and the continuous deformation results in the substructures consisting of entangled dislocations in dynamically recrystallized grains. (ii) More severe shear deformation at a large strain rate intensifies dynamic recrystallization to a certain extent. (iii) At a large strain rate, most of the deformation heat cannot be dissipated, instead, it is stored as heat energy, leading to the temperature rise of specimen.

The deduction should be accurate, and there is a poor connection to the obtained results. Also, the relationship between strain rate and mechanisms of recrystallization should be explained in a logical way with references.

Author Response

1- The abstract is not appropriate, and not representative of the main results. In the abstract, the results should be summarized briefly. 

Our response:

Thanks, we have revised abstract.

2- The English language of the manuscript is misleading, and the manuscript is not easy to read for readers. It is recommended to be revised by an English native speaker who is an expert in the field.

Our response:

 Thanks, we will invite an English native expert to revise the language before publishing.

3- The introduction should be extended. I could not understand why this study is essential. The motivation should be explained after reviewing the previous related works.

Our response:

Thanks. We have extended introduction.

Compared to hot forming, the warm forming need less energy to heat the materials to lower forming temperature, but the warm rolling torque was 3 times of hot rolling torque(P2.Ref.29). So, there's controversial whether total energy savings in warm rolling. Moreover, the spheroidization of cementite and the recrystallization of ferrite had a remarkable effect on the tensile strength and yield strength.

Therefore, the investigation of the temperature sensitivity of the material,the influence of process parameters on microstructure and texture evolution, and the distribution of dislocation density reflecting the homogeneity of plastic deformation, is helpful and essential to optimize technical parameters (temperature, strain rate, area reduction , etc.) to accomplish the better forming qualities with fewer resources(Ref.31).

4- In line 88, the definition of LAGB and HAGB were stated. This needs to add references: https://doi.org/10.1016/j.ijhydene.2020.12.028

https://doi.org/10.2320/matertrans.45.3247

Our response:

 Thanks, we have added the two references. They are helpful.

5- There are several typos and errors in writing. The manuscript should be proofread before submission!

Our response:

Thanks, we have proofread the manuscript.

6- In line 140, the authors explained that dynamic recrystallization (DRX) of ferrite was hard to occur and dynamic recovery (DRV) was the main softening mechanism, due to the high stacking fault energy of BCC structure which was susceptible to dislocation climb and cross slide. What is Cross slide? The right one is cross slip. Please explain why cross slip and climb are easier in high SFE materials.

Our response:

Yes, the correct expression is cross slip, and the corresponding word has been revised in the manuscript (P.5). According to previous reports, the width of extended dislocation is small in high SFE materials, which commonly leads to the clustered imperfect dislocations. During heat deformation, dislocation climb and cross slip easily proceed, resulting in sufficient dynamic recovery. However, the remaining stored energy is insufficient to facilitate the dynamic recrystallization. We have added some sentences and a reference to discuss this issue(P. 5 and Ref.36).

7- In line 168, three reasons were mentioned for the different extent of recrystallized grains in different strain rates.  (i) The deformation time is longer at a low strain rate and the continuous deformation results in the substructures consisting of entangled dislocations in dynamically recrystallized grains. (ii) More severe shear deformation at a large strain rate intensifies dynamic recrystallization to a certain extent. (iii) At a large strain rate, most of the deformation heat cannot be dissipated, instead, it is stored as heat energy, leading to the temperature rise of specimen.

          The deduction should be accurate, and there is a poor connection to the obtained results. Also, the relationship between strain rate and mechanisms of recrystallization should be explained in a logical way with references.

Our response:

In the above mentioned part, we discussed the relationship between the strain rate and recrystallization behavior. The corresponding two references for these deductions have been added in the revised manuscript (P.7 and Ref. 38,39).

Reviewer 2 Report

-Please check that the formatting is in line with the MDPI template - specifically the references do not seem to be in the right format - see for instance line 27 "rolling1" - I take it this is "rolling [1]. I will just continue revieweing assuming that you will fix the formatting.
-Your study is particularly relevant for diffusion bonding - a process during which you experience considerable deformation at elevated temperatures (https://doi.org/10.3390/met10050613) - this can be mentioned in the introduction
-how is warm forming an energy saving technology? compared to hot forming? cold forming should save most energy...need better explain this
-the study is justified and the state of the art is explained
-Explain better how the gleeble can be used to simulate the condition of material in an oven - I think gleeble uses electricity to heat the sample - not just radiation - please explain why this does not affect the results/preductions of your study about samples heated by other means
-How did the quenching happen - did you drop the samples in water (at room temperature)?
-are there trade names for the materials that you used - if so please name them
-Fig 8 seems to be cut short please check
-the results are fine - presentation especially with regards to formatting needs to be improved

Author Response

-Please check that the formatting is in line with the MDPI template - specifically the references do not seem to be in the right format - see for instance line 27 "rolling1" - I take it this is "rolling [1]. I will just continue revieweing assuming that you will fix the formatting.

Our response:

Thanks , we have fixed the formatting according to the MDPI template.

-Your study is particularly relevant for diffusion bonding - a process during which you experience considerable deformation at elevated temperatures (https://doi.org/10.3390/met10050613) - this can be mentioned in the introduction

Our response:

Thanks, we have added the reference(P.2,Ref.31), it is very helpful.

-how is warm forming an energy saving technology? compared to hot forming? cold forming should save most energy...need better explain this

Our response:

Compared to hot forming, the warm forming need less energy to heat the materials to lower forming temperature, but the warm rolling torque was 3 times of hot rolling torque(P2.Ref.29). So, there's controversial whether total energy savings in warm rolling , that is also one of the purposes of this paper.

-the study is justified and the state of the art is explained

Thanks!

-Explain better how the gleeble can be used to simulate the condition of material in an oven - I think gleeble uses electricity to heat the sample - not just radiation - please explain why this does not affect the results/preductions of your study about samples heated by other means

Our response:

Final temperature and heat balance of the materials are the two main factors influencing the microstructure. Each sample was heated to the deformation temperature and held for 3 minutes under isothermal conditions. The sample is so small and is easy to reach heat balance. Therefore, As long as the temperature is stable and balanced, no matter what heating method is used, the result will not be affected.

-How did the quenching happen - did you drop the samples in water (at room temperature)?

Our response:

Yes, The sample with size ofφ8×12mm is small and can be immediately quenched in water.

-are there trade names for the materials that you used - if so please name them

Our response:

It is 20CrMoA(P.2).

-Fig 8 seems to be cut short please check

Our response:

Thanks ,we have corrected the figure.

-the results are fine - presentation especially with regards to formatting needs to be improved

Our response:

Thanks , we have fixed the formatting. we will invite an English native expert to revise the language before publishing.

Round 2

Reviewer 1 Report

The revised version seems better that the previous version. However, the English language of the manuscript needs improvements.

Reviewer 2 Report

All comments have been addressed - I will recommend the publication of this manuscript.